# Inflammatory Bowel Diseases: An Updated Overview on the Heat Shock Protein Involvement

**DOI:** 10.3390/ijms241512129

**Published:** 2023-07-28

**Authors:** Federica Scalia, Francesco Carini, Sabrina David, Marco Giammanco, Margherita Mazzola, Francesca Rappa, Noemi Irma Bressan, Giorgio Maida, Giovanni Tomasello

**Affiliations:** 1Biomedicine, Neurosciences and Advanced Diagnostics BIND, School of Medicine, University of Palermo, 90133 Palermo, Italy; francesco.carini@unipa.it (F.C.); margheritamazzola@hotmail.it (M.M.); francesca.rappa@unipa.it (F.R.); giorgio.maida@community.unipa.it (G.M.); giovanni.tomasello@unipa.it (G.T.); 2Euro-Mediterranean Institute of Science and Technology (IEMEST), 90139 Palermo, Italy; 3Hospital University School of Medicine, P. Giaccone, 90127 Palermo, Italy; 4Department Surgical, Oncological and Oral Sciences, School of Medicine, University of Palermo, 90133 Palermo, Italy; sabrina.david@unipa.it (S.D.); marco.giammanco@unipa.it (M.G.); 5Institute of Translational Pharmacology (IFT), Section of Palermo, Italy National Research Council of Italy (CNR), 90146 Palermo, Italy; 6School of Medicine, University of Palermo, 90133 Palermo, Italy; noemiirma.bressan@gmail.com

**Keywords:** IBD, molecular chaperones, immune, HSPs, Crohn’s disease, ulcerative colitis

## Abstract

Inflammatory bowel diseases (IBDs) represent chronic idiopathic disorders, including Crohn’s disease (CD) and ulcerative colitis (UC), in which one of the trigger factors is represented by aberrant immune interactions between the intestinal epithelium and the intestinal microbiota. The involvement of heat shock proteins (HSPs) as etiological and pathogenetic factors is becoming of increasing interest. HSPs were found to be differentially expressed in the intestinal tissues and sera of patients with CD and UC. It has been shown that HSPs can play a dual role in the disease, depending on the stage of progression. They can support the inflammatory and fibrosis process, but they can also act as protective factors during disease progression or before the onset of one of the worst complications of IBD, colorectal cancer. Furthermore, HSPs are able to mediate the interaction between the intestinal microbiota and intestinal epithelial cells. In this work, we discuss the involvement of HSPs in IBD considering their genetic, epigenetic, immune and molecular roles, referring to the most recent works present in the literature. With our review, we want to shed light on the importance of further exploring the role of HSPs, or even better, the role of the molecular chaperone system (CS), in IBD: various molecules of the CS including HSPs may have diagnostic, prognostic and therapeutic potential, promoting the creation of new drugs that could overcome the side-effects of the therapies currently used.

## 1. Introduction

Inflammatory bowel diseases (IBDs), including Crohn’s disease (CD) and ulcerative colitis (UC), represent chronic idiopathic diseases, whose etiopathogenesis involves many different factors belonging to the host (human body) and to the guest (gut microbiota) [1]; they are: genetic predisposition, epigenetic variations, environmental factors, alterations of the intestinal microbiota, deregulation of the immune response and chronic inflammation.

All these factors are indisputably linked to each other. For example, gene variations, as well as epigenetic modifications caused by environmental factors such as stress, diet, breastfeeding, infections, cigarette smoking, vaccinations, use of antibiotics and many others [2,3], can cause the alterations of genes and proteins with the role of controlling the immune response and, subsequently, can lead to chronic inflammation [4]. However, it would be correct to take into consideration two controversial aspects of IBD immunity: (I) the presence in the mucosa and serum of patients with UC of autoantibodies against, for example, intestinal epithelial cells (antiGP-2, GAB), perinuclear neutrophilic proteins, tropomyosin isoform 5, neutrophil cytoplasmatic proteinase PR-3 and pancreatic cells and the presence of anti-CD99 antibodies demonstrate the autoimmune feature of IBDs [5,6,7], but this aspect has to be distinguished from (II) the immune-mediated issue of IBDs. Indeed, it mainly happens when abnormalities of the cell-mediated immune system lead to the loss of tolerance of the system toward the intestinal microbiota attacking not the body cells but the pathogenic intestinal bacteria, infectious agents (e.g., viruses) or food in the gut, causing inflammation that leads to bowel injury [8,9].

In addition, molecular mimicry phenomena should not be excluded in IBDs since microbial peptides may have structural similarities to self-peptides and may induce the activation of autoreactivity [8].

Therefore, despite the plethora of factors that gravitate around these disorders, the term IBD refers only to the inflammatory condition of the lining tissue of the digestive tract. However, while the entire gastrointestinal tract is involved in CD, the last tract of the intestine, comprising the colon and rectum, is affected in UC. Nevertheless, symptoms may be similar in CD and UC, such as diarrhea, stomach pain, rectal bleeding, fatigue and weight loss [10,11].

In addition to the characteristics of the host (genetics, epigenetics, immunity), the microbiota status is another interesting point of view for understanding the pathogenic processes underlying IBDs. Research in the field of IBD has focused a lot on the interactions between the intestinal microbiota, genetic background and the immune system, to try to understand the genetic-molecular mechanisms underlying these diseases and find therapeutic targets. The gut microbiota is one of the most important trigger factors in the development of IBD, as aberrant interactions between the gut epithelium and the microbiota can provoke inflammatory responses [12]. It seems that the host cells of the lining gut mucosa, with their genetic background, are able to shape the microbiota composition. This dependence between host genetics and gut microbiota composition has been demonstrated both in human and animal models: monozygotic twins showed a more similar microbiome than dizygotic twins [13], and mice model knockouts of IBD-related genes showed significant changes in gut microbiota [1]. In turn, the composition of the microbiota, and in particular its beneficial microbes, is pivotal for the homeostasis of immune responses, locally and systemically, and for the maintenance of mucosal tolerance to inflammatory processes [14]. In this regard, the microbiota of subjects affected by IBD is characterized by a reduced microbial diversity and an imbalance in favor of pro-inflammatory bacteria and against anti-inflammatory ones [15]. Furthermore, the integrity of the intestinal mucosa depends on the activation of the immune system which can be regulated by the intestinal microbiota [16]. Therefore, the microbiota is not simply an element that populates our gastrointestinal system, but it also collaborates and regulates activities such as nutrition, metabolism and immune response and protects against pathogens [17]. 

The number of studies interested in the association between IBD and heat shock proteins (HSPs) is growing considerably. HSPs are well-conserved proteins expressed by the cells in stress conditions, such as heat shock (from which they take their name) oxidative and chemical stress, exposure to UV light and autoimmune and chronic inflammation [18]. Under stress conditions, the heat shock response (HSR) pathway is activated and initiates the activation of the transcription factor (HSF) which is able to bind the heat shock regulatory element (HSE) of the DNA to initiate the transcription of several downstream molecular chaperons’ genes. Six heat shock factors (HSF1,2,3,4, HSFX and HSFY) exist, but HSF3 is only expressed in chickens and mice. HSF1 is the most studied heat shock factor able to regulate several physiological and pathological processes within the cell, including inflammation, apoptosis and immune processes, and can play a role alone or in combination with HSF2 [19]. However, some of the HSPs have been found also constitutively expressed, or expressed at a low level, by the cell in non-stress conditions and may be upregulated when cells need protection. The genetic or acquired dysregulation of HSPs is related to several pathological conditions called chaperonopathies [20,21], many of which are characterized by inflammatory processes and autoimmune disorders which can affect different organs and systems including the gastrointestinal system [22,23]. HSPs are classified according to their molecular weight into six families (HSP110, HSP90, HSP70, HSP60 and HSP10, also called chaperonins, HSP40 and small HSPs) [24]. They have the chaperone role to promote the folding of nascent proteins or the refolding of the misfolded proteins, and they act as disaggregases against protein precipitates and aggregates and promote the assembly of multiprotein complexes. In addition, they activate the ubiquitin–proteosome pathway to eliminate the dangerous damaged proteins from the cells [24]. This role appears fundamental for IBD if we think that Paneth cells, special granule-containing cells found in the epithelial crypts of the small intestine, play an essential role in innate intestinal defenses, and it has been shown that dysbiosis resulting from the misfolding of alpha-defensin in Paneth cells can contribute to the pathogenesis of CD [25]. Furthermore, HSPs also have an immunomodulatory role, especially when they are secreted into the extracellular space and activate immune cells by presenting a peptide to cells with MHC of I and II class [26]. Heat shock gene mutations and deregulations have been associated with IBD and with complications associated with this condition such as colorectal cancer (CRC) and intestinal fibrostenosis [27,28]. The molecular mechanisms of IBD have long remained unknown. HSPs appear to play a role in many of the pathophysiological processes associated with the onset and progression of IBDs. In this review, we discuss the involvement of HSPs in IBDs according to their genetic condition, molecular contribution and immunological role. 

## 2. Heat Shock Protein Expression in Healthy Gastrointestinal Mucosa and IBD Mucosa

In general, a large amount of HSPs is expressed by cells comprising the mucosa of the large intestine. Their expression is normally upregulated following stimuli provided by intestinal bacteria flora or after the fermentation of digested food [18]. So far, research has demonstrated that HSPs have a cytoprotective role in gastric mucosa; however, the lack of knowledge on the chaperone system (CS), intended as the network comprising molecular chaperones and their interactors, regulators, client proteins and others (which is therefore much more complex than the single study on molecular chaperones), does not allow one to identify one complete pathway in which the molecular chaperones are involved.

That HSPs can play a protective role for the mucosa of subjects affected by IBD is demonstrated by the high expression of the first factors of the HSR cascade, i.e., HSF1 and HSF2. HSF2 has been shown overexpressed in the mucosa and sera of patients with UC, and, moreover, the expression correlates with the severity of the pathology [29,30]; HSF1 together with HSP70 has been demonstrated to be protective during the induction of colitis or gastric lesions in mice [31].

However, the cytoprotective role of HSPs in intestinal mucosa is controversial and has always been questioned. It has been hypothesized that HSPs may have a protective role or they may play a role on behalf of inflammatory bowel disorders. It should be clarified to specifically determine whether they are valuable IBD targets of treatment.

Nevertheless, the beneficial role of sets or individual HSPs was shown in several research studies: the oral administration of anti-ulcer drugs induced HSP60, HSP70, HSC70 and HSP90 expression in the cell mucosa of rats ameliorating the pathological phenotype [32].

HSP70 is the most studied HSP observed in UC and CD. Increased levels of HSP70 have been related to a stronger resistance of mucosa to damage caused by the repeated administration of aspirin [33], and in animal models, HSP70 has been demonstrated to ameliorate the progression of atrophic gastritis [27]. Furthermore, recent research demonstrates through HSPA1A knockdown in the 18Co-cell line and Caco-2 cell line that the absence of HSP72 is related to the epithelial–mesenchymal transition of intestinal epithelial cells demonstrating its involvement in intestinal fibrostenosis and in the maintenance of healthy intestinal mucosa [34]

However, together with HSP27, HSP70 is predominantly localized to the surface of epithelial cells, and therefore, it has been suggested that they may regulate the immune response in IBD; for example, HSP70 may induce a pro-inflammatory response through toll-like receptor-2 and toll-like receptor-4. [35,36]. Also, HSP60 and HSP10 were found overexpressed in the colon mucosa of patients with UC, and, mainly in the CD tissues of patients, the level of HSP60 decreased after treatment with 5-aminosalicylic acid (5-ASA) [37]. On the other hand, Hsp90 has been detected expressed equally in IBD and healthy patients [38]. However, HSP90 was found higher in the epithelium but not in the lamina propria of UC mucosa; as reported for HSP60, the expression of HSP90 in the lamina propria along with HSP10 (the co-chaperone of HSP60) and HSP70 has been shown to be diminished through immunohistochemistry following treatment with 5-aminosalicylic acid (5-ASA) [39]. Finally, HSP47, a chaperone expressed in the endoplasmic reticulum and involved in the biogenesis, folding and secretion of collagens, has been evaluated expressed at a higher level in the intestinal tissues of patients with active/inflamed CD [40].

## 3. Genetic Alterations in IBD- and *HSP*-Related Genes

The genetic basis of IBD has been increasingly studied in recent years, so much so that a database has been established, the International IBD Genetics Consortium (https://www.ibdgenetics.org/ (accessed on 20 April 2023)), which studies and evaluates the genetic differences in patients with IBD [41].

Through genome-wide association analysis (GWAS), researchers demonstrate the polygenic nature of IBDs and the presence of several genetic polymorphisms [42]. The GWAS data identified 240 loci associated with IBD [43], some of which specific for CD and UC, responsible for the clinical, endoscopic and histological differences between the two pathologies [44]. Most loci are associated with both pathologies, suggesting that these disorders share common inflammatory pathways, as shown in Table 1 [42,43]. Furthermore, in accordance with the inflammatory nature of IBD, GWAS studies identified numerous T-helper (Th) 17, Treg and interleukin-related IBD susceptibility genes, including *JAK2, STAT3, IL-23R, IL-12B* and *CCR6* [45].

The belief that a deep genotyping of the patient’s tissues could provide decisive information for choosing the right therapeutic approach is becoming increasingly strong.

Among polymorphisms some *HSP* genes seem to be involved in IBD susceptibility; however, the studies reported to date demonstrate that the presence of polymorphisms related to CD or UC are strictly dependent on the ethnicity of the population under consideration, and some examples are here illustrated.

The HSP70 family genes (HSP70-1, -2 and -Hom) are localized in the short arm of the human chromosome number six, close to the gene encoding for the human leukocyte antigen (*HLA).* The genotype of HSP70 has been related both to cancer and to a reduced risk of the gastric premalignant condition [46], demonstrating that the polymorphisms may be decisive for the protective or non-protective role of the encoded protein. The PstI polymorphism of the *HSP70-2* gene (allele A) was found associated with CD susceptibility in Chinese people; meanwhile, the allele B of the same gene has been reported associated with the perforating form of CD in the population of Japan and America but not in Tunisia’s population [47,48,49].

The same assessment was performed for the nucleotide-binding oligomerization domain 2 (*NOD-2*) gene, also known as *CARD-15* (caspase activation and recruitment domain 15), the first CD susceptibility gene identified [1]. It is an intracellular receptor containing pattern-recognition receptors (PRRs) able to recognize pieces of the pathogen’s wall (muramyl dipeptide (MDP)), in the context of pathogen-associated molecular patterns (PAMPs). This recognition triggers the pro-inflammatory pathway leading to the activation of the NF-κB molecule. Patients with variants of the *NOD2* gene are characterized by dysfunction in bacteria recognition; uncontrolled inflammation occurs, and CD may develop with a higher probability [50]. *NOD2* risk variants are associated with lower levels of α-defensin in Paneth cells leading to reduced antimicrobial function [51]. The prolonged intestinal inflammation is supported by studies in *Nod2*−/− mice in response to bacterial infections [52]. The *NOD2* gene polymorphism was associated with CD in the USA and Western Europe but was not reported to be correlated with CD in Tunisian, Japanese, Chinese and Korean people [53]. Of great interest for the purpose of this review is the relationship between *NOD2* and the molecular chaperones. *NOD2* is regulated by several molecules within the cell: the Erbin protein is able to inhibit the ability of *NOD2* to interact with bacterial fragments; HSP90 binds to *NOD2* before its activation, and following the presence of MDP, HSP90 and *NOD2* dissociate [54]. These mechanisms allow the mediation of an excessive recognition of *NOD2* toward the bacterial flora. Also, HSP70 was found bound to *NOD2*. HSP70 has been demonstrated to be able to stabilize the wild-type *NOD2* pathway; in fact, its overexpression is correlated to a greater activity of NF-kB, but it is also able to allow the recovery of the *NOD2* CD-associated variant allowing *NOD2* to improve the recognition of bacterial MDP [54]. 

To date, no polymorphisms of the gene coding for HSP90 have been reported associated with IBD; however, a polymorphism of this chaperone is associated with the response to treatment with glucocorticoids in patients affected by the systemic lupus erythematosus (SLE), an autoimmune disease [55]. Different polymorphisms may increase susceptibility to IBD or may act as protection factors, around the world (Figure 1). In turn, it demonstrates that the development of IBD depends on environmental factors meeting a specific genetic background.

Therefore, the study of the polymorphisms of HSPs, in particular HSP70 and HSP90 (but also HSP60 and its co-chaperone HSP10 found increased in the mucosa of patients with IBD [37]), appears to be crucial for understanding the molecular pathways underlying the pathogenesis of IBD and their heritability in different countries.

## 4. Epigenetic Alterations in IBD- and HSP-Related Proteins 

Epigenetic alterations of the DNA sequence such as chromosomal instability (CIN), microsatellite instability (MSI), hypermethylation and defects of noncoding RNA have been reported in inflamed tissue before the diagnosis of colitis-associated cancers (CACs), the worst outcome of IBD [56]. The CIN epigenetic alteration is the most common alteration from which an IBD can turn into a CAC and may cause aneuploidy with the consequent loss of the gene (e.g., *p53*, *APC*, *IDH1*, *Rb* and *ARID1A*) and/or activation of proto-oncogenes (e.g., *K-ras*,*28 C-src*,*29 C-myc*,*25* and *PIK3CA24*) [56]. 

Epigenetic alterations are strictly dependent on environmental factors, including the diet (to which also the intestinal microbiota is closely linked), and can be inherited: monozygous twins with different environmental histories show diverse epigenetic imprinting [57] and diverse microbiota. 

Some research studies demonstrate that the microbiota can alter the epigenetic status of genes [58]. In brief, microbiota-derived metabolites, such as the short-chain fatty acids (SCFAs) produced by the microbial fermentation of dietary carbohydrates or the S-adenosylmethionine (SAM) generated during the folate–methionine cycle performed by several commensal bacteria like the *Bifidobacterium* and *Lactobacillus* species, provide donors as methyl and acetyl, crucial for the catalytic activity of the regulator enzymes of the epigenetic events (e.g., methylases, acetylases, histone deacetylases, histone acyltransferases and methyltransferase) which can occur in the interfaced-to-bacteria intestinal epithelial cells (IECs) [59,60,61]. Methylation or hypermethylation causing gene silencing is a typical epigenetic alteration in IBDs, including UC-, CD- and IBD-associated neoplastic lesions [56]. In recent years, the analysis of the circulating epigenome in IBD is becoming of great interest. Methylation changes in the HLA region and *VMP1/MIR21* have been identified in Scottish children with CD [56], and, on the other hand, in adult IBD patients, in the *RPS6KA2* and *VMP1* and *ITGB2* and *TXK* genes, respectively, differentially methylated positions (DMPs) and differentially methylated regions (DMRs) have been demonstrated [62]. In particular, the DMRs *VMP1, ITGB2, WDR8* and *CDC42BPB* have been discovered associated with CD, and the DMRs *VMP1* and *WDR8* have been related to UC [61]. Of interest appears to be the methylation of the 3′-end region of the *VMP1* gene, both in children and adults, since the *VMP1* gene shares its region with the primary transcription site for microRNA-21 (pre-miR21), a microRNA implicated in the inflammatory state in colitis and IBD [63].

It has been demonstrated that a methyl-deficient diet is related to colitis progression and may induce the acetylation of *HSF1* causing a dramatic decrease in chaperones under its regulation such as the binding immunoglobulin protein (BIP), HSP27 and HSP90 [64] (Figure 2).

A close contribution of methylation in IBD, and the study on the IBD-specific circulating methylome [65], demonstrates that not only genetic mutations but also epigenetic alterations should be considered in the research for the best therapeutic IBD approach. 

Furthermore, the use of drugs may alter the epigenome of IECs (and the microbiota) and may be beneficial or not for the IBD patient. Many of the drugs used, or those under investigation, to treat IBD belong to Chinese medicine and seem to have epigenetically mediated anti-inflammatory action during the progression of the IBD [65]. An example is Triptolide, a bioactive molecule, extracted from *Tripterygium wilfordii* Hook F, able to regulate the HSP70 pathway during the migration of fibroblasts [66].

This highlights that the set-up of the microbiota in patients with IBD should also be considered when a therapeutic microbial-based approach is applied, since it could somehow alter the commensal bacterial populations in the intestinal environment and in turn the epigenetic of the IECs and heat shock proteins.

## 5. Immunology and Inflammation in IBD and HSP Involvement

Alongside the genetic-molecular mechanisms, there are immunological mechanisms associated with the pathogenesis of IBD. Intestinal immune cells can be divided into innate immune cells (macrophages, dendritic cells, neutrophils, natural killer cells, innate lymphoid cells) and adaptive immune cells, both of which contribute significantly to immune responses in IBD [67]. The inflammation is triggered by the interaction between innate cells which are then able to produce cytokines, chemokines and antimicrobial agents. It leads to phagocytosis, antigen presentation and the activation of the adaptive immune system [68].

It has been reported that, from the immune point of view, CD is mediated by Th1/Th17, while UC is associated with a Th2-type response [67]. 

Immunological mechanisms are also responsible for the development of fibrosis in IBD, a serious complication affecting approximately 30% of patients with CD and 5% of patients with UC [67]. Intestinal fibrosis is characterized by chronic, recurrent or unresolved intestinal inflammation, which contributes to the excessive accumulation of the extracellular matrix (ECM) and the loss of normal function, with the consequent risk of intestinal obstruction [67]. An important immune role in the pathogenesis of IBD is represented by interleukins such as IL-12, IL-23, IL-10 [67], IL-17A, IL-17F, IL-21 and IL-22 produced by the T-helper (Th) 17 cells involved in the inflammation and fibrosis of tissues. 

The role of HSPs in the inflammation of IBD is a topic of great interest for investigation. The factors HSF1 and HSF2 and the heat shock proteins HSP90, HSP70, HSP60 and HSP27 have been considered in various works in the literature; therefore, for the involvement of these heat shock proteins, we suggest the work still present in the literature [19,23,69] while we deepen, below, the involvement of other molecules. Honzawa and collaborators demonstrated an upregulation of HSP47, the chaperone of collagen, in intestinal active/inflamed tissues of patients with CD which correlated with a higher level of IL-17A and IL-22. IL-17A has been demonstrated to be able to induce HSP47 expression [40]. Many CD patients undergo intestinal structuring, one of the major complications of CD, which is caused by a recurrent inflammation of intestinal mucosa associated with immune cell infiltration and epithelial damage. In turn, this process is followed by a strong deposition of collagen contributing to fibrotic stricture formation [70]. Furthermore, Kurumi and colleagues have recently demonstrated a higher level of HSP47 and anti-HSP47 antibody in the serum of CD patients than UC patients [70]; therefore, HSP47 may be considered a potential candidate for the treatment of fibrotic CD which should be preferred to the inhibitor of transforming growth factor-β-1 (TGF-β1) used so far. TGF-β and its receptors have a critical role in the development of intestinal fibrosis; in fact, they are particularly overexpressed in intestinal fibro-stenotic CD cells and in animal models of intestinal fibrosis. The activation of the TGF-β signaling pathway leads to extracellular matrix production and inhibits the expression of matrix metalloproteinases (MMPs) [71]. Blocking the TGF-β signaling pathway can reduce intestinal fibrosis [67]; however, its use is controversial due to its potent immunosuppressive effect on intestinal inflammation [67,72]. 

Alpha B-crystallin, also called CRYAB or HSPB5, is a small heat shock protein that prevents the aggregation of proteins in many tissues such as heart, skeletal muscle and brain. Its deregulation or disturbance is particularly studied in neurodegeneration, neuroinflammation and neuromuscular disorders [73,74], and when it is located in the extracellular space, it is also able to modulate the response to inflammation [75]. In IBD, Xu and collaborators demonstrate that CRYAB was decreased in the inflamed mucosa of IBD patients, and it seems to be activated by a proinflammatory cytokine TNF-α. CRYAB binds to IKKβ and inhibits its activity which is required for the activation of the NF-κB pathway [76]. In turn, TNF-α synergizing with IFN-γ kills intestinal epithelial cells and disrupts intestinal epithelial barrier function via CASP8-JAK1/2-STAT1 [77] and through the activation of NF-κB signaling. TNF-α is a very present marker in IBD and is closely associated with intestinal fibrosis through the promotion of myofibroblast proliferation and collagen accumulation [67]. Therefore, TNF-α is also involved in the fibrotic process promoting the activation, proliferation and accumulation of collagen in fibroblasts [67].

Moreover, Cryab−/− mice are more likely to develop autoimmune and inflammatory disorders [78]. The role of alpha B-crystallin in the regulation of inflammation is still to be investigated, but even in this case, a heat shock protein could become a new therapeutic target for the treatment of IBD and fibrosis-associated IBD.

Finally, concerning autoimmunity in IBD, we should mention the presence of autoantibodies against HSP40 and HSP60 in IBD patients [79,80]. However, even if the role of autoantibodies against HSP60 has to be investigated, their presence may be correlated to the high homology with its microbial counterparts; instead, the origin of anti-HSP40 is controversial, and, interestingly, it has been shown to be beneficial in UC patients promoting a slowdown of the disease [81].

## 6. Involvement of HSPs in the Progression of IBD to Cancer

The need to investigate the pathogenesis of IBDs so that targeted/personalized and efficient therapeutic actions can be generated is also necessary to counteract the onset of colitis-associated cancers (CACs), the most serious complication of IBDs. To date, a genetic pattern associated with IBD capable of establishing a minor or major susceptibility to the onset of colon cancer has not been identified; therefore, this very serious event cannot be monitored or predicted. Analyzing the gene mutations in lesions associated with CAC, a clear genomic field related to cancer risk has not been identified [82]. Furthermore, the independence of CAC from the Wnt pathway, a typical molecular pathway activated in sporadic colorectal cancer (CRC) [82], demonstrates that IBD-dependent CAC should be studied as a standalone lesion. Nongenetic but epigenetic events, drivers of genetic alterations, may be induced by chronic inflammation and may lead to the increased risk of developing CAC. 

HSPs have been discovered to be overexpressed in a wide range of human cancers and are involved in almost all tumorigenic processes [82]. Despite the protective role of the alpha B-crystallin discussed in the previous section, overexpression of alpha B-crystallin has been associated with several forms of cancer, including CRC [83]. HSPB5 was demonstrated to be a potent inducer of epithelial–mesenchymal transition, the central mechanism during cancer invasion, transformation and metastasis [84]. In addition, its overexpression was associated with a poor prognosis in CRC patients suggesting its prognostic role during the follow-up of patients [84]. Similarly, HSF1, which was demonstrated to prevent colitis in mice through the activation of HSP70 [31], seems to lose its protective role during the progression of inflammatory intestinal disorders playing a carcinogenic role within the cells [85] and when it is located in the extracellular matrix during the pre-malignant inflammatory stage; here, HSF1 promotes ECM remodeling [86]. 

## 7. Conclusions

With this review, we wanted to highlight the genetic, epigenetic, molecular and immune mechanisms in which HSPs are involved that underlie the development of IBD. Understanding these mechanisms is of fundamental importance in order to develop increasingly targeted therapies that take into account molecular chaperones. Identifying *HSP* susceptibility genes in different populations from around the world may contribute to monitoring the expansion and heritability of high-IBD-susceptibility polymorphisms; in addition, the study of epigenetic modifications and microbiota composition in patients could pave the way for targeted and personalized therapies. Understanding immunological mechanisms triggered or protected by HSPs may allow for the development of even more personalized therapeutic strategies for immunotherapy or microbial-based therapy.

We do not exclude that other information on the involvement of HSPs in IBD exists in the literature, but the multifactorial nature and complexity of these disorders often hide the role of HSPs. Furthermore, HSPs are often referred to by a different nomenclature that has no reference to their “heat shock” condition; for example, HSPA9 is also known as mortalin/mtHSP70/GRP75/PBP74; the HSP40 family is also called DNAJ, and in turn, DNAJC28 is also known as Orf28 open reading frame 28/C21orf55/oculomedin; HSPB8 is also called H11/HMN2/CMT2L/DHMN2/E2IG1/HMN2A/HSP22; HSPC4 also has the name of HSP90B1/ECGP/GP9/TRA1/GRP94/endoplasmin [87].

However, we referred to the works that cite these molecules according to the most used “HSP” nomenclature.

## Figures and Tables

**Figure 1 ijms-24-12129-f001:**
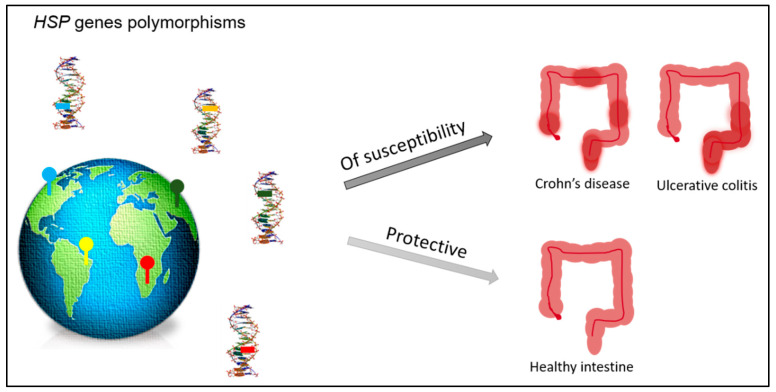
Polymorphisms of the genes encoding for HSPs may occur in different countries. Some of them may act as factors of susceptibility to the development of IBD (Crohn’s disease or ulcerative colitis), while others may act as protective factors assisting in the maintenance of the healthy/non-inflamed intestinal tissue.

**Figure 2 ijms-24-12129-f002:**
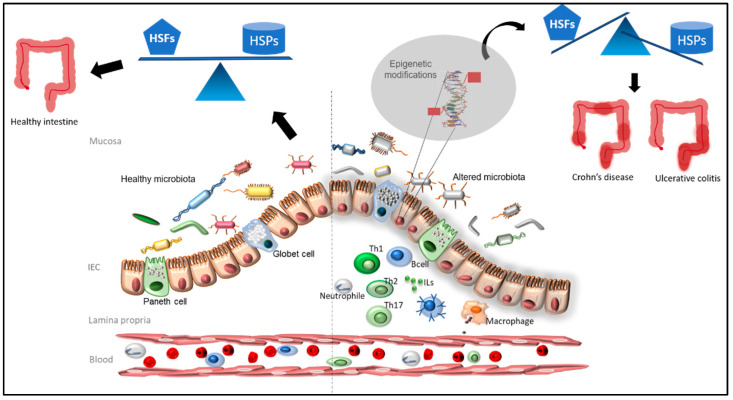
The healthy and balanced microbiota (on the left) is able to provide the optimal metabolites for the maintenance of the epigenome of healthy intestinal epithelial cells (IECs). When microbiota is altered (on the right), and/or an inflammation process occurs, the intestinal flora is not able to provide the proper microbiota-derived metabolites inducing aberrant epigenetic modifications of HSP genes in IECs. If deregulation occurs for the genes encoding for the heat shock factors (HSFs), the expression of the heat shock proteins (HSPs) under HSF control is lost, and an IBD may occur.

**Table 1 ijms-24-12129-t001:** Some genes involved in IBD pathogenesis [43] (adapted from Ref. [43] Copyright © 2023 Elsevier Ltd. All rights reserved.).

Role	IBD	CD	UC
IL23/T_H_17 pathway	*IL23R, JAK2, TYK2, ICOSLG, TNFSF15*	*STAT3*	*IL21*
Autophagy	*CUL2*	*ATG16L1, IRGM, NOD2, LRRK2*	*PARK7, DAP*
T-cell regulation	*TNFSF8, IL12B, IL23, PRDM1, ICOSLG*	*NDFIP1, TAGAP, IL2R*	*tNFRSF9, PIM3, IL/R, TNFSF8, IGNG, IL23*
Innate mucosal defense	*CARD 9, RER*	*NOD2, ITLN1*	*SLC11A1, FGR2a/B*

## Data Availability

Not applicable.

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
