# Peer review of "Inflammatory Bowel Diseases: An Updated Overview on the Heat Shock Protein Involvement"

_ijms, 2023, doi:10.3390/ijms241512129_

Round 1
Reviewer 1 Report
This review aims to examine the molecular mechanisms to date studied in IBD. The idea has merit but the presentation from a linguistic point of view in many parts in the manuscript is very critical and should be revised with editing of the English by a native speaker.
Some considerations:
MAJOR:
- There is no space in this review, except in a few, poorly detailed parts, for the role of the gut microbiota, which has extremely relevant pathogenic and immunologic implications of IBD. Certainly a new section should be added for this;
- it is particularly a limitation of this review that it does not present any figures. It is a deficiency that should be corrected as dealing with this review of innumerable molecular pathways the use of figures to help the reader fix these mechanisms is imperative;
- Most of the studies in this review are quite dated. What does this review add to large narrative reviews already produced? Especially to particularly important reviews already published (Ex. doi: 10.1056/NEJMra2002697)?
- In addition, another big issue is that of RNAs in the pathogenesis of IBD such as small noncoding RNAs, microRNAs: there are over 600 studies on the subject and the authors mention only pre-miR21;
- Certainly even if it is beyond the scope of such a review, how can one introduce topics such as JAK-STAT, IL-22, IL-23 and not describe (or at least mention to reinforce the pathogenetic role of such pathways) tofacitinib, upadacitinib, ustekinumab, etc.?
- Also no mention is made of the role of the melanocortin system on which more than 200 papers have been produced;
- No mention is made of the sphingosine-1-phosphate pathway on which many of the new therapies are being based...
- There is no mention on future perspective, pilot studies done and similar...
MINORS:
- Line 56: correct "Crohn's disease."
- Line 57: the acronym "Crohn's Disease (CD)" exposed above is repeated again, also from the definition take out "chronic" IBDs already incorporate chronicity in their definition;
- Line 57: I would not say it affects the entire intestine but I would say "potentially may affect the entire gastro-intestinal tract." This definition is incorrect;
- Line 59: this can simply be written as "remitting-recurrent course."
- Line 61: Crohn's disease again instead of CD. Why "correct diagnosis". Just say diagnosis is enough.
- Line 74: I would say "typical" not "usually" transmural;
- Check all acronyms well, even in the UC paragraph acronyms are re-presented;
- Line 87: what do you mean by "continuous fashion"?;
Ultimately I think this review should be greatly expanded and corrected for further evaluation. I expect to see new studies, more novel mechanisms for a narrative review on IBD pathogenesis in 2023.
Moderate english revision required
Author Response
We thank the reviewer for their very helpful comments and suggestions which we have followed up in preparing the revised manuscript now presented. The work was heavily modified and we focused on the involvement of heat shock proteins in IBD. Focused on HSPs involvement, we tried to answer to reviewer's requests:
- we referred to the microbiota and autoimmunity.
- we added figures
- we have included recent works and new studies in the field of HSPs in IBD
The corrections to the lines are not visible as the work has been heavily edited, the sections have changed and much of the text has been deleted.
We hope the manuscript is now acceptable for publication.
Reviewer 2 Report
This paper highlights the genetic, epigenetic, molecular and immune mechanisms that underlie the development of IBD. The article shows genetic information obtained through genome-wide association studies that identified 240 loci associated with inflammatory bowel disease.
English language in this paper is good enough
Author Response
We thank the reviewer for the comments. The work was heavily modified and we focused on the involvement of heat shock proteins in IBD.
The corrections are not visible as the work has been heavily edited, the sections have changed and much of the text has been deleted.
We hope the manuscript is now acceptable for publication.
Reviewer 3 Report
The authors have provided a review article aiming to present an overview of the molecular and immune mechanisms involved in inflammatory bowel diseases (IBD). IBD refers to a group of chronic inflammatory conditions that primarily affect the gastrointestinal tract, including Crohn's disease and ulcerative colitis.
- It is recommended to prepare a Graphical summery of the suggested mechanisms of IBDs
- The review requires additional improvements to include the most recent studies. For instance, the authors did not mention anything about autoimmunity in IBDs and the presence of autoantibodies in IBDs!
The quality of the written text needs to be improved. For instance: the following sentences represent poor scientific writing.
“From an immune point 16 of view, however, inflammatory bowel diseases see the involvement of Th1/Th17 and Th2 responses.”
“The need to investigate, genetically and molecularly, the pathogenesis of IBDs in order that targeted/personalized and efficient therapeutic actions can be generated is also necessary to counteract the onset of colitis-associated cancers (CACs), the most serious complication of IBDs “
Author Response
We thank the reviewer for the comments and suggestions which we have followed up in preparing the revised manuscript now presented. The work was heavily modified and we focused on the involvement of heat shock proteins in IBD. Focused on HSPs involvement, we tried to answer to reviewer's requests:
- we referred to the microbiota and autoimmunity.
- we added figures
- we have included recent works and new studies in the field of HSPs in IBD
The corrections to the lines are not visible as the work has been heavily edited, the sections have changed and much of the text has been deleted.
We hope the manuscript is now acceptable for publication.
Reviewer 4 Report
Dear Authors,
Overall, this review is not very well organized, focused, and sometimes hard to read. What is the main aim addressed by the review? The abstract is poorly written and does not reflect the most important points of the paper. The introduction should be qualified with more supporting evidence (only 10 references were used). The novelty/importance of this review is very weak. What does this review add to the subject area compared with other published reviews? A clear description of the evidence gap that this review is filling is needed. The method section is not reported. Authors reviewed studies but it is not clear how these were chosen for inclusion. The search strategy should include search terms, study type (e.g., in vivo/in vitro), databases, inclusion and exclusion criteria...etc. Genes associated with IBD should be described in much more details. I miss figures to provide a visual overview of the key message(s) of immunological mechanisms in IBD. Are the conclusions consistent with the evidence presented and do they address the main aim posed? A list of abbreviations should be included at the end of the paper.
Moderate editing of English language required.
Author Response
We thank the reviewer for the comments and suggestions which we have followed up in preparing the revised manuscript now presented. The work was heavily modified and we focused on the involvement of heat shock proteins in IBD. Focused on HSPs involvement, we tried to answer to reviewer's requests:
- we clarified the aim of the review
- we added figures and references
- we have included recent works and new studies in the field of HSPs in IBD
The corrections are not visible as the work has been heavily edited, the sections have changed and much of the text has been deleted.
We hope the manuscript is now acceptable for publication.
Round 2
Reviewer 1 Report
Revisions were done.
Author Response
We thank the reviewer for the attention given to our manuscript. Text changes are visible in track changes.
Reviewer 4 Report
Dear Authors,
The paper has significantly improved by these revisions. Few points remain:
1. Please delete bullet points in introduction.
2. The method section is missing. Several studies were reviewed but it is not clear how these were chosen for inclusion in the review. The search strategy should include search terms, study type (e.g., in vivo/in vitro), databases, inclusion and exclusion criteria...etc.
3. In section 4, I would suggest adding more details on genetic and epigentics of IBD. There are several reviews on this topic. In addition, please expand on how SCFAs as key epigenetic metabolites could play a role in the treatment of IBD. Please refer to these articles (EBioMedicine. 2021 Apr; 66: 103293; Nutrients. 2022 Oct 3;14(19):4113).
4. Is this review without limitations? I would suggest adding a few lines of any limitations raised.
N/A
Author Response
We thank the reviewer for helpful comments and for the attention given to our manuscript. Text changes are visible in track changes.
1. We did what the reviewer asked and eliminated the bulleted list.
2. We thank the reviewer but for this kind of review the methods section is not required by the journal, however we have included something about it in the conclusions.
3. We thank the reviewer but this request would take us away from the focus of this review, i.e. the contribution of HSPs in IBD, and may confuse the reader. We could soon write a review only on the topic requested by the reviewer, as it is very broad.